# The Use of Armeo^®^Spring Device to Assess the Effect of Trunk Stabilization Exercises on the Functional Capabilities of the Upper Limb—An Observational Study of Patients after Stroke

**DOI:** 10.3390/s22124336

**Published:** 2022-06-08

**Authors:** Anna Olczak, Aleksandra Truszczyńska-Baszak, Adam Stępień

**Affiliations:** 1Military Institute of Medicine, Rehabilitation Clinic, 128 Szaserów Street, 04-141 Warsaw, Poland; 2Faculty of Rehabilitation, Józef Piłsudski University of Physical Education in Warsaw, 00-968 Warsaw, Poland; aleksandra.truszczynska@awf.edu.pl; 3Military Institute of Medicine, Neurological Clinic, 128 Szaserów Street, 04-141 Warsaw, Poland; astepien@wim.mil.pl

**Keywords:** Armeo^®^Spring device, ischemic stroke, upper limb, motor coordination, trunk stabilization exercises, NDT Bobath concept

## Abstract

Almost half of stroke patients report impaired function of the upper limb and hand. Stability of the trunk is required for the proper movement of the body, including the legs and arms. The aim of the study was to analyze the effect of trunk stabilization exercises on coordinated movement of the affected upper limb in patients after stroke, using an Armoe^®^Spring device and the “wall” and “abacus” functional tests. Materials and methods: This is a randomized, double-blinded study. The research was carried out in the Rehabilitation Clinic on a group of 60 stroke patients who were randomly assigned to groups differing in the rehabilitation program. The study group had physiotherapy based on the NDT Bobath concept and the control group used classic exercises. The importance of the trunk for upper limb coordination was assessed on the Armeo^®^Spring device using three evaluation programs, “perpendicular fishing”; “horizontal fishing”; “reaction time”, and two proprietary tests, “wall” and “abacus”. Results: The post-treatment analysis showed significantly better results in the study group for the abacus (*p* < 0.001), wall (*p* = 0.003) tests, and a significantly higher percentage of task completion in the vertical fishing (*p* = 0.036) and reaction time (*p* = 0.009) tests. Conclusions: Physiotherapy including exercises to stabilize the trunk had a significant effect on increasing the functional efficiency of the affected upper limb and on improving the handgrip strength. The Armeo^®^Spring device is a good device for the functional assessment of the upper limb before and after therapy.

## 1. Introduction

Stroke is a very common disease in the human population, and the most common consequences of the disease are disability, vascular dementia, depression and epilepsy [1]. The acute phase of stroke in most patients is characterized by a decline in the efficiency of the upper and lower extremities and the trunk [2,3]. In the chronic phase, more than half of the post-stroke patients still have a loss of mobility in the upper limb [4]. Moreover, the regeneration of the upper limb usually takes longer than that of the lower limb, which may be due to the short time of unit therapy of the upper limb. As reported by Hayward et al., the arm training lasted from 4 to 5.7 min and from 23 to 32 repetitions per session during physical therapy, as was the case in a hospital setting, and slightly longer, from 11 to a maximum of 17 min per session, during occupational therapy [5]. The researchers point out that after a stroke, the level of bilateral motor coordination in the upper limbs decreases [6,7,8,9,10,11,12,13]. Many researchers maintain that the trunk plays a very important role in the functioning of the body. It plays a key role in the control of posture and in the performance of selective movements of the trunk and limbs [14,15,16,17,18].

The concept of a stable trunk was already dealt with by Peto. Its concept, with reference to the development of an infant, presents and formulates the stability of the trunk as an integrated and harmonious work of the postural flexors and extensors [19]. It is an indispensable model for all activities, the basic pattern of movement. Alhwoaimel reported that trunk exercises improve trunk performance in people with acute, subacute and chronic strokes and that there is no evidence that trunk exercises affect upper limb function [20]. The trunk counterbalances the movements of all limbs. If it works properly, it is possible to make selective movements around the body circumference. It is very important that the trunk is both stable and mobile, because only then can any selective movement be made [21]. Trunk control is important for functioning and may also be an early predictor of daily activities [16,17].

In turn, Haruyama et al. investigated the effectiveness of trunk stabilization training for trunk function, balance in standing, and mobility in stroke patients. They found that trunk stabilization training had a beneficial effect on the trunk function and balance in standing positions and mobility in patients after a stroke [22]. The results of the stability studies indicate that in patients after strokes, trunk stabilization exercises should be introduced [23,24,25,26]. Lee et al. assessed the influence of breathing exercises on trunk control, assuming that the trunk muscles play a large role in the work of the respiratory system [27]. On the other hand, Kim and the above-mentioned Hariharasudhan, Park, and their teams used the proprioceptive neuromuscular priming method in their work with patients after stroke, and in each case showed a positive effect on postural control [28,29,30]. The NDT Bobath concept also appeared in reports on the possibility of influencing muscle quality and trunk control, central stabilization, gait, and daily activities [31]. Desouzart wrote about the effectiveness of the NDT Bobath concept in improving motor skills parameters in children with cerebral palsy, especially in terms of general motor skills, postural control, and stability [32]. In addition, Keser and those mentioned emphasized the importance of NDT Bobath in working with patients with multiple sclerosis [33]. On the one hand, in studies comparing the NDT Bobath concept with other exercises, the described concept plays a greater role in the possibility of trunk stabilization [34]. On the other hand, when comparing the NDT Bobath concept and the PNF method in working with stroke patients, in order to improve the motor control of the trunk, both work options turned out to be effective [35]. However, after analyzing the literature, we did not find any work in which the authors would show the effect of trunk stabilization exercises, according to the NDT Bobath concept, on the coordination of the upper limbs. Therefore, in our work, we present the impact of this concept on the functional capabilities of the upper limb in patients after stroke. For the functional assessment of the upper limb, we decided to use the Armeo^®^Spring device. Armeo^®^Spring is a modern device from the Armeo^®^ family for diagnosing ranges of motion and coordination of the upper limb and is also used in neurorehabilitation. Physiotherapy consists of working with an orthosis (exoskeleton), the spring system of which supports the rehabilitated upper limb and supports training. The brace is intended for patients with limited or lost shoulder functionality [36]. The use of Armeo^®^Spring for upper limb training allows you to increase muscle strength, range of motion and motor skills of the upper limb. Colomer et al. presented the use of this device in the improvement of the upper limb in patients after a stroke at an early stage [37]. Armeo^®^Spring was used by Gueye et al. to work with elderly patients after a stroke [38]. Gijbels et al. described the use of Armeo^®^Spring in the physiotherapy of children with cerebral palsy [39]. In turn, Adomaviˇciene et al. presented a comparison of the applications of Armeo^®^Spring and Kinetic Kontrol [40]. It is difficult to find any reports on the use of this device for examining the upper limb. Therefore, in our work, this device was used for functional evaluation of the affected upper limb in patients after stroke.

The aim of the study was to assess the impact of trunk stabilization exercises on the functional capabilities of the upper limb using the Armeo^®^Spring device and the “wall” and “abacus” functional tests.

## 2. Methods

### 2.1. Trial Design

The trial was a randomized, double-blinded study that lasted 10 days. The patients were divided into the following two groups: study and control and were subjected to different therapies (independent variables). The study group consisted of 30 people. Their physiotherapy was based on exercises that heavily employed the core muscles to equalize tension and strength, according to the NDT Bobath concept. The control group also consisted of 30 patients. They underwent classical neurological rehabilitation. All patients were examined twice, the first time was after admission to the Rehabilitation Clinic, and the second time was after 10 days of therapy. Assessment games were used for the study, which involved the software of the Armeo^®^Spring device and proprietary tests, “wall” and “abacus” (dependent variables).

The criteria for stroke group inclusion were as follows: (1) patients with ischemic stroke; (2) patients with hemiparesis after 5 to 7 weeks after stroke; (3) subjects with poor trunk control (the trunk control test 48–61 points); (4) subjects who were in a functional state allowing movements of the upper extremity (FMA-UE 43–49 motor function points); (5) muscle tension (MAS 0–1+); (6) no severe deficits in communication, memory, or understanding of what can impede proper measurement performance; (7) at least 35 years of age; maximum 85 years of age.

The criteria for stroke group exclusion were as follows: (1) lack of possibility to adjust the orthosis to the patient’s treated limb, (2) bone instability (non-fused fractures, advanced osteoporosis), (3) permanent contracture of the treated limb, (4) open skin lesions in the area of the treated upper limb, (5) sensory deficits disturbances, (6) shoulder subluxation or pain (7) increased spasticity, (8) increased involuntary movements, e.g., ataxia, dyskinesia, myoclonic seizures, (9) unstable life functions, including contraindications related to the respiratory system or the cardiovascular system (instability or the need to use supportive devices), (10) the need for long-term intravenous therapy, (11) postural instability, (12) contraindication to a sitting position, (13) confused or uncooperative patients, (14) severe cognitive impairment, (15) patients requiring isolation due to infections, (16) severe vision problems (the patient is not able to observe the elements displayed on the computer screen), (17) epilepsy.

### 2.2. Therapeutic Intervention

The research was carried out according to the protocol no 4/KRN/2020, registered in Clinical Trial Registration.

Patients had physiotherapy at the Rehabilitation Clinic 6 days a week (Monday to Saturday). Patients were randomly allocated to Bobath or classical physiotherapy.

The procedure in the study group was based on the use of the NDT Bobath concept, the aim of which is to influence the development of the trunk control skills. Exercises in closed kinematic chains also play an important role. For example, the support on the directly affected upper limb obtained in different starting positions causes pressure on the articular surfaces (approximation), which in turn improves propioception, increases the stabilization of the shoulder joint and increases muscle tension (Appendix A). The exercises were carried out in various starting positions, including lying on the back, on the sides, lying on the front, sitting, as well as in supported kneeling, straight kneeling and standing. Therapy based on the NDT Bobath concept improves everyday functioning and the patient’s body structures (tension, strength, ranges of motion, etc.) by teaching activity, rotation, sitting down, getting up and practicing various variants of walking. Additionally, exercises for tilting the trunk in the sagittal plane and forwards and sideways in the sitting and standing positions are intended to integrate the body parts and improve the equivalent responses by establishing the body midline Appendix A).

On the other hand, exercise at the activity level, reaching a designated point on the front left and right with both hands, crossing the centerline of the body with trunk rotation, can improve the function of the stabilizing muscles of the trunk. Similarly, standing exercises, walking in place with the pressure on the ball with both arms activates the abdominal muscles and improves central stabilization. Moreover, it facilitates alternating and selective work of the lower limbs (Appendix A).

Treatment in the control group was based on the use of classic exercises, such as passive exercises, but also approximation. In addition, the patients performed self-assisted exercises on a manual rotor, and in order to relieve the directly affected limb, the patients exercised in a suspension system. As the physiotherapy progressed, the patients performed active exercises, and then active exercises with resistance, e.g., using the Thera Band. Patients also exercised their balance with the use of large gymnastic balls or sensor pads. A large part of the physiotherapy was locomotion training and gait re-education.

The duration of the treatment session for each patient in both groups was 120 min.

### 2.3. Devices and Tests Used at Work

A ARMEO^®^SPRING device was used (Hocoma AG, Volketswil, Switzerland). The main element of the device is an orthosis (exoskeleton), which has a system of springs supporting the exercised upper limb (Figure 1). The design of the device allows for the adjusting of the orthosis to the patient. The adaptability of the device to the patient is ensured by the electrically adjustable column in the range of 400 mm, the length of the forearm in the range of 290–390 mm, the length of the arm in the range of 220–310 mm, the maximum weight of the forearm from 0.7 kg to 2.4 kg, the maximum weight of the arm from 0.5 kg to 3.8 kg. The Armeo Spring has 6 degrees of freedom (each with an independent motor and two sensors), thanks to which angular movement are possible in the range of adduction/abduction in the shoulder joint −169° to +50°, flexion/extension in the shoulder joint +40° to +120°, internal/external shoulder rotation 0° to 90°, elbow flexion/extension 0° to 100°, forearm pronation/supination −60° to 60°; wrist flexion/extension −60° to 60°. In addition, the device has a pressure sensor for the grip. The measuring accuracy of the device is <0.2 degree according to the manufacturer.

The Armeo^®^Spring software enables the creation of patient databases, individualization of therapy parameters for each patient, modifies the levels of difficulty of exercises, includes games and tasks to motivate the patient, and provides transparent reporting. Armeo^®^Spring is a professional tool for assessing the progress of therapy. It has three diagnostic programs called evaluation games, including “vertical fishing”, “horizontal fishing”, and “reaction time” [36].

The movement is initiated by the patient and the main advantage of the device is the detection of the position and movement of the arm. Moreover, the support of the arm makes it possible to increase mobility in 3D space. An important feature of the device is also biofeedback (Figure 2).

### 2.4. Evaluation Games

#### 2.4.1. Vertical Fishing

The patient’s task is to catch a ladybug. In this evaluation game, the patient has to move his hand in a vertical plane. When it touches a ladybug, the ladybug disappears and a new one appears elsewhere. If the patient does not touch the ladybug within the allotted time, the ladybug also disappears and another reappears elsewhere.

#### 2.4.2. Horizontal Fishing

As in the previous game, the patient’s task is to catch the red ball. In this evaluation game, the patient has to move his upper limb horizontally. The rules for the appearance and disappearance of an item are the same as in the previous game.

In both games, the patients were tested on the 1st level of difficulty, which means that the field of work was 40 × 30 cm^2^ and they had 12 objects to catch.

#### 2.4.3. Reaction Time

The patient moved his hand in the frontal plane to catch the fly. The rules of the game are the same as in the previous games, with the difference that each time the patient has to return to the center of the screen to the shelf and should remain on the shelf until another object appears on the screen. As in the previous games, the study was on level I, working area 30 × 26 cm^2^ and the patients had to catch 20 objects.

Apart from the Armeo Spring device, two proprietary tests were used to examine the patients, “abacus” and “wall”.

“WALL” TEST. It consists of lifting the upper limb on one’s own and moving the limb along the wall as high as possible. Patients performed it sitting straight with their knees pressed against the wall. Patients were scored according to the following schedule:The patient does not raise the upper limb;The patient raises the upper limb to shoulder height;The patient raises the upper limb to the height of the head;The patient raises the upper limb above the head.

The purpose of the test was to assess the functional capabilities in the shoulder joint of the affected upper limb. This test examined whether the patient could overcome the force of gravity.

The “ABACUS” TEST was performed on classic mathematical abacuses. The patient’s task was to move the beads from one edge of the abacus to the other with two fingers (index and thumb) of the affected upper limb. The result of the test was the number of beads moved in 30 s. The test assessed the grasping activity of the upper limb (precise grip, according to Napier, pincer or paddle grip, apical grip) [41].

### 2.5. Examination Procedure

Before starting the therapy and after 10 days of therapy, the patients were examined once in the following assumed order: the wall test, the abacus test, and then three evaluation games on the Armeo Spring. The patient was seated during all tests. The use of evaluation games on the Armeo Spring requires, first of all, the individual adjustment of the orthosis to the patient’s upper limb. Before using the device, the physiotherapist checked that there were no severe contractions in the muscles of the upper limb, and that the patient was properly dressed to minimize the risk of skin damage and irritation. Then, the shoulder joint was blocked with two pins. The patient was sitting in a chair or in a wheelchair, while the ArmeoSpring was positioned behind the patient.

#### Armeo^®^Spring Assessment Parameters

The evaluation parameter used in the “vertical fishing” and “horizontal fishing” games is the hand movement path coefficient. It is used to assess the patient’s quality of movement. This parameter was calculated by the quotient of the length of the trajectory of the patient’s hand movement to the distance between the points that can be achieved in individual elements of the “vertical fishing” and “horizontal fishing” exercises. This ratio showed the extent to which the patient deviates from the shortest straight line connecting two objects when moving from one object to another. If the movement is perfect (the shortest in a straight line), the hand movement path coefficient is 1. If the coefficient is 3, it means that the patient’s hand movement trajectory is three times longer than the shortest line that connects two objects.

The evaluation parameter used in the ‘reaction time’ game is the measurement of the time taken for the patient to react. The software of the Armeo^®^Spring device measures the time from the moment the first object (a fly) appears on the screen to the moment it leaves the shelf, i.e., the starting base. Then, the time is measured from the moment the fly disappears to the moment it returns to the shelf-base [36].

### 2.6. Ethics

Regarding ethical approval, the study was carried out in the Rehabilitation Clinic of the Military Medical Institute (MMI) in Warsaw, Poland. It was approved by and carried out by the recommendations of the Ethical Committee of the Military Medical Institute (MMI); approval number 4/MMI/2020. Before inclusion, all subjects were informed about the purpose of the study. Written informed consent was obtained from all subjects by the tenets of the Declaration of Helsinki.

### 2.7. Sample Size Calculation

The sample size was estimated using the G * Power 3.1.9.4 program. Assuming the following parameters of effect size d = 0.59, α = 0.05; power = 0.8 for the Wilcoxon–Mann–Whitney test, the required sample size is 76 (38 people per group). Assuming that these parameters were met in the analysis, the adopted sample was sufficient.

### 2.8. Statistical Analysis

Statistical analyzes were performed using IBM SPSS Statistics 26.0. In order to compare the two groups, an analysis was performed with the Mann–Whitney U test. In order to compare the two measurements, analysis was performed with the Wilcoxon test. The level of significance was α = 0.05.

## 3. Results

### 3.1. Participants

In total, 80 stroke patients were examined. After the exclusion criteria, 20 people were excluded because of the period of the disease (5 people), their functional condition (10 people), and some refused to participate (5 people). The National Institute for Health Stroke Scale (NIHSS) [42] was used to identify the neurological deficit, and to evaluate the patients’ overall physical impairment. The FMA UE test was used to assess the functional state of the upper limb to assess the stability of the trunk and the trunk control test (TCT), and to assess the tension of the muscle, the Ashworth modified scale (MAS) was used [43,44,45]. The flow of participants through each stage of the study is shown below (Figure 3).

Sixty patients after ischemic cerebral stroke (men and women, average age 65.83 ± 10.40) were randomly recruited from among patients of the Rehabilitation Department of the MMI.

Patients were in the acute phase of the disease (5–7 weeks post-stroke), with slight neurological deficits (NIHSS ≤ 7).

The stabilization of the trunk (from 48 to 61 points in TCT, functional state of the upper limb) and enabling movements were measured (FMA EU from 43–49 motor function points, and normal sensation/light touch. The tension of the muscles was measured with the Modified Ashworth Scale (MAS 1/1+). The clinical evaluation of patients after a stroke was performed by the physician admitting the patient to the clinic on the day of admission. The characteristics of the patients were shown in Table 1 and Table 2.

Finally, 60 patients were randomly divided into the study group (30 people) and the control group (30 people). In the study group, exercises to stabilize the trunk were used in accordance with the NDT Bobath concept, while in the control group, classic exercises were used.

### 3.2. Outcomes and Estimation

The results of our work can be analyzed in the order consistent with the following research questions:Did the test patients differ significantly before therapy?How did each of the therapy affect the analyzed results?Does an exercise stabilizing the trunk may change the functional state of the upper limb, in patients after stroke, essentially?

#### 3.2.1. Compare the Results of the Study and Control Groups for the Pre-Treatment Measurement

In order to compare the studied groups of patients before the exercises, the Mann–Whitney U test compared the results of people randomly assigned to the study and control groups.

The conducted analysis showed that in the control group, they obtained a significantly higher result for the vertical and horizontal fishing time, while in the study group, they achieved a higher percentage of task completion (reaction time). For the remaining variables, the differences between the groups for the pre-exercise measurement were statistically insignificant. The results are presented in Table 3.

#### 3.2.2. Compare the Results before and after the Exercises in Each of the Studied Groups

Study group. The conducted analysis showed significant differences between the measurements for most of the analyzed variables. In the measurement after stabilization exercises, higher results were obtained in the tests abacus and wall, vertical fishing (% execution), horizontal fishing (% execution), and a lower result for vertical fishing time, reaction time, and hand movement path coefficient for vertical fishing. There were no differences between the measurements for horizontal fishing-time, path coefficient and % of task completion for reaction time. The results of the analysis are presented in Table 4.

Control group. The analysis showed higher results for the second measurement in the abacus, wall, vertical, and horizontal fishing tests (% execution), and lower values for vertical and horizontal fishing execution time and path coefficient, as well as reaction time-time. The results of the analysis are presented in Table 5.

#### 3.2.3. Comparison of the Results of the Study and Control Groups after Therapy

The analysis was performed using the Mann–Whitney U test.

For the measurement after exercise, the analysis showed higher results in the study group for the abacus, wall, vertical fishing (% execution), and reaction time (% execution) tests. A lower result, compared to the control group, was obtained for the time of vertical fishing. For the remaining variables, the differences between the groups turned out to be insignificant. The results of the analysis are presented in Table 6.

## 4. Discussion

The results of our research show that physiotherapy of patients after a stroke, taking into account exercises improving the stability of the trunk, has a statistically significant impact on the functional improvement of the affected upper limb, as well as on the increase in the grasping ability of the hand. Trunk stabilization exercises were carried out in accordance with the NDT Bobath concept. The assumption of this concept is to achieve the greatest possible independence of the patient in performing everyday activities [46].

The study of the function of the upper limb in our work was carried out, inter alia, using the Armeo^®^Spring device, which, in addition to the evaluation games, contains many games for everyday training of the upper limbs.

Schwarz et al. presented in their work a qualitative analysis of upper limb movement in the chronic period of the disease after stroke in patients with mild or moderate mobility impairment of the upper limbs. They confirmed that the kinematic measures of mutual coordination in the shoulder-elbow-trunk syndrome largely depend on the motor task and the examined arm [47]. In turn, the usefulness of devices similar to Armeo was tested by Lee, et al. They tested the effectiveness and therapeutic usefulness of virtual canoeing training for stroke patients. Therapeutic efficacy was assessed on the basis of trunk stability, balance, and motor function of the upper limb. The results showed that this type of training is an acceptable and effective therapeutic method, improving all the above-mentioned study parameters [48]. The results of the study by Liao et al. found that healthy people had greater movement speed when flexing their arms, flexing and extending the trunk. In the study group, a greater lateral shift of the center of body mass was demonstrated during the movements of all limbs and trunk. Moreover, studies have shown that any activity in the limbs and trunk disturbs the stability of the trunk in stroke patients [49]. In turn, the results of the research by Hodge and Richardson confirmed that exercises stabilizing the trunk by working on the deep abdominal muscles more effectively increase the range and smoothness of limb movement than other forms of exercise [50]. Similarly, other researchers confirmed the importance of the tension of the muscles that deeply stabilize the trunk for the work of the lower limbs and coordinated movements of the trunk [51].

At the beginning of our study, we checked whether the studied groups of patients before the therapy were similar in terms of functional evaluation. We obtained the result that the study group had a higher percentage of task completion in the reaction time test, while the control group obtained higher scores in the two tests, horizontal and vertical fishing time. After 10 days of therapy, both in the study group and in the control group, the patients obtained many statistically significant results. Among them, the results of the wall and abacus tests, as well as horizontal and vertical fishing (percentage of task completion), were significantly higher in both groups after the therapy. Moreover, in the control group, after the therapy, much better results were obtained in the reaction time test. Therefore, analyzing each of the groups separately and the influence of the therapy on the functional efficiency of the upper limb, slightly higher results were obtained in the group dominated by the therapy based on traditional physiotherapy. However, when comparing the results of both groups after the procedures, it turned out that in the study group in which the stabilizing exercises were performed, according to the NDT Bobath concept, the patients obtained more statistically significant results. The results of the abacus and wall trials turned out to be higher, as did the percentage of the task completion in vertical fishing trials, as well as the reaction time. For comparison, in the control group, the significant results concerned only the execution time in the vertical fishing test.

In this way, our research confirmed that the stabilization exercises according to NDT Bobath are necessary for the functional improvement of the affected upper limb. We obtained not only an improvement in the mobility of the upper limb but also an improvement in the precise pincer grip. Moreover, it is worth recalling that our research was conducted on stroke patients in the acute phase of stroke.

Similarly, Lee et al. examined 46 patients, a minimum of 6 months after the stroke. He divides the patients into two groups, the first group exercised the upper limbs with symmetrical contraction of the abdominal muscles, and the second group exercised without the abdominal muscles. They found that there was a significant improvement in balance in the study group. Based on their research, the authors recommend trunk stabilization exercises as part of post-hospital exercises in patients after stroke [52]. In turn, researchers Beer et al. demonstrated in their works that the ability to straighten the elbow after a stroke depends on the size and direction of the torques acting on the shoulder [53].

The results of our work, obtained on the Armeo^®^Spring device, showed that trunk stabilization exercises in accordance with the NDT Bobath concept improve the mobility and coordination of the affected upper limb in stroke patients in the acute phase of the disease.

Armoe^®^Spring is a device that enables movement in all joints of the upper limb. The adjustable support system makes the task easier or more difficult. This allows you to accurately direct the movement of the upper limb, which helps in practicing the correct movement patterns of the upper limb. Many researchers have used this device for therapy at different stages of stroke, but the Armeo^®^Spring device can also facilitate functional analysis of the upper limb. The test results showing the percentage of motor task completion or the time of motor task completion as well as the movement path coefficient allow for the conclusion of the motor abilities in all planes, as well as present differences in the motor coordination of the upper limb. The Armeo^®^Spring device can be used for therapy and diagnostics of the upper limb, which is confirmed by the results of our work.

### 4.1. Research Value

Physiotherapy of the patient after stroke, including exercises aimed at improving the stability of the trunk, has a significant impact on increasing the functional efficiency of the affected upper limb and on improving the handgrip strength and should be included in the rehabilitation program of patients after a stroke.

### 4.2. Study Limitation

The limitations in our work that may affect the objectivity of the results are a smaller group of test persons, according to the sample calculation, in addition to the relatively short time of therapy between the first and second examinations. Moreover, taking into account the purpose of the work, we believe that the methodology of measurements on the Armeo Spring device may also be a limitation of this study. The instructions for use assume that the patient is in a sitting position with back support. This support acts as stabilization for the trunk and our goal was to check how the NDT Bobath trunk stabilization exercises change the functionality of the upper limb. It seems that it would be better to examine patients without support before and after therapy or to perform the examination twice before and twice after the applied therapies (with and without back support). Subsequent studies will take into account the above-mentioned limitations of the current study.

## 5. Conclusions

The patient’s physiotherapy, including exercises aimed at improving the stability of the trunk, has a significant impact on increasing the functional efficiency of the affected upper limb and on improving the handgrip in terms of pinch grip.

The Armeo^®^Spring device is a good device for assessing the functional evaluation of the upper limb before and after therapy.

Exercise to improve trunk stability should be included in the rehabilitation program for stroke patients.

## Figures and Tables

**Figure 1 sensors-22-04336-f001:**
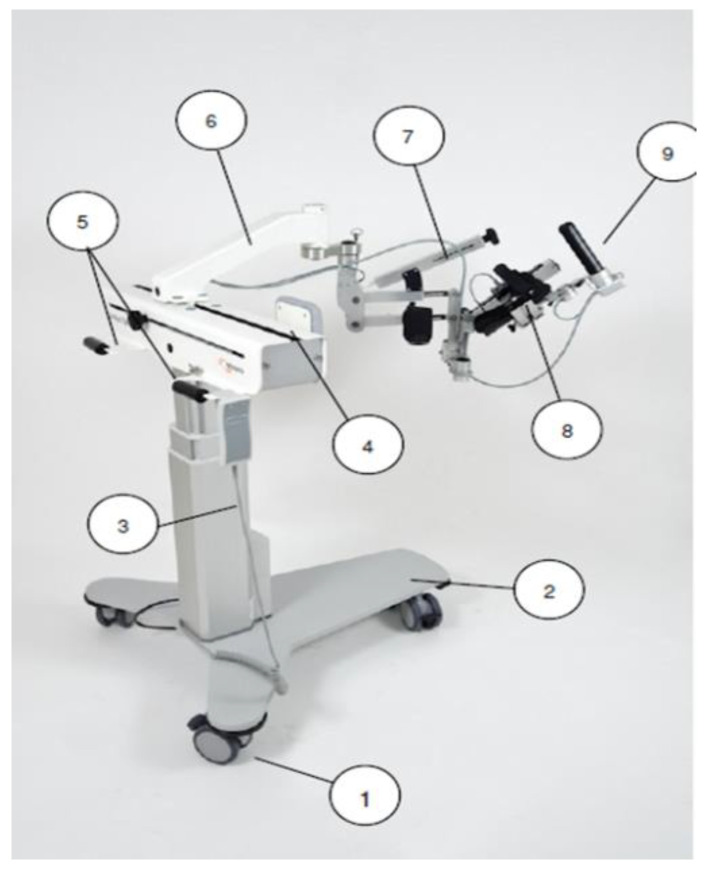
Armeo^®^Spring Overview. 1—wheel, 2—base plate, 3—lifting column, 4—rope guide, 5—transport handles, 6—sliding arm with integrated electronics, 7—arm module with weight compensation mechanism and cuff, 8—forearm module with mechanism weight compensation and circular guide for integrated pronation and supination, 9—pressure-sensitive handle (source: Armeo^®^Spring User Manual, Hocoma AG).

**Figure 2 sensors-22-04336-f002:**
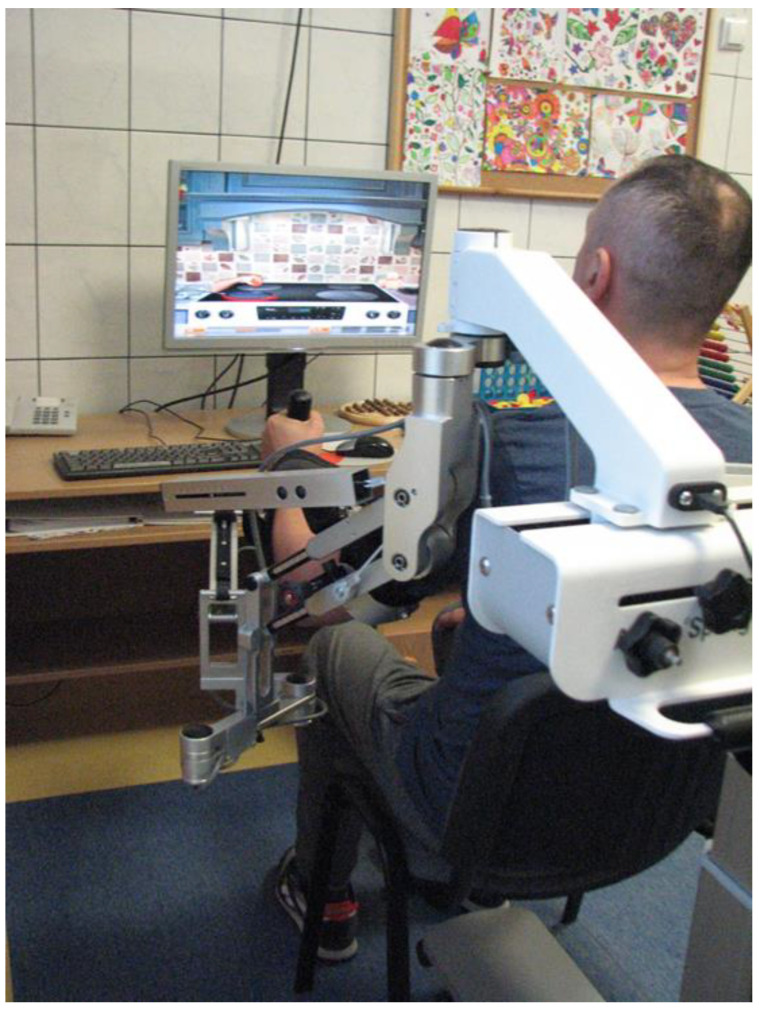
Armeo Spring device. Rearview. Visible coupling of movements performed by the patient depending on the task given in the program visible on the computer monitor (source: own collections).

**Figure 3 sensors-22-04336-f003:**
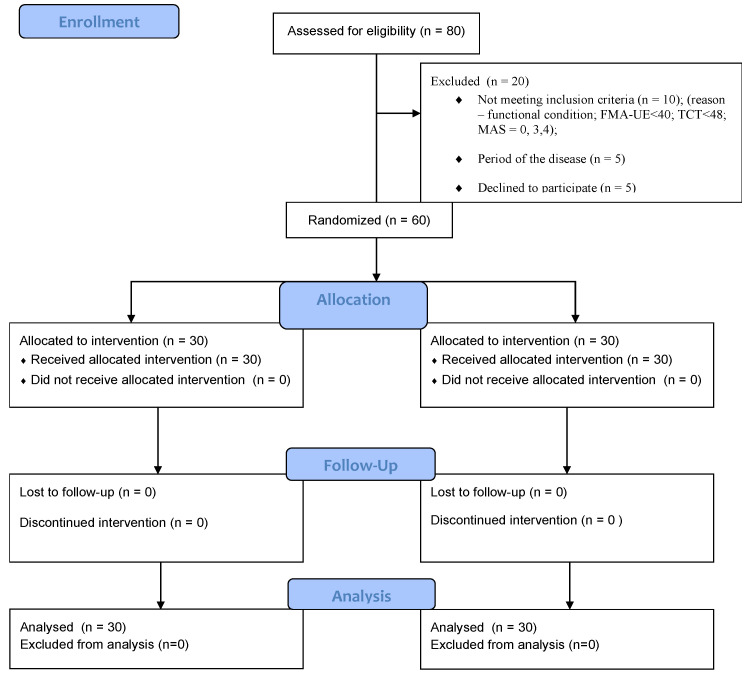
The flow of participants through each stage of the study.

**Table 1 sensors-22-04336-t001:** Demographic characteristics of stoke participants.

Post-Stroke Group	Study	Control
Age (years) mean ± SD	65.27 ± 10.56	66.40 ± 10.40
Height (cm) mean ± SD	166.77 ± 8.39	168.67 ± 7.81
Weight (kg) mean ± SD	79.77 ± 13.09	78.80 ± 12.82

**Table 2 sensors-22-04336-t002:** The basic epidemiological data of the study and control stroke participants.

Post-Stroke Groups	Study	Control
n = 60 (100%)	30 (50%)	30 (50%)
Female	15 (50%)	15 (50%)
Male	15 (50%)	15 (50%)
Cerebral ischemic stroke (thromboembolic) n/%	30 (100%)	30 (100%)
Time post stroke/episode (weeks)	5–7	5–7
Right affected side	15 (50%)	15 (50%)
Left affected side	15 (50%)	15 (50%)
Dominant right hand	28 (93.3%)	28 (93.3%)
Dominant left hand	2 (6.67%)	2 (6.67%)
TCT (points 48–61) ± SD	53.20 ± 6.31	53.63 ± 6.47
FMA-UE (points 43–49) ± SD	45.47 ± 1.87	45.50 ± 2.11
MAS (degrees 0/1/1+) (examined n)	0/1/1+ 0/20/10	0/1/1+ 0/20/10

**Table 3 sensors-22-04336-t003:** Comparison of people from the study and control groups for the measurement before the exercises.

Before	Study (n = 30)			Control (n = 30)					
	Average Rank	Me	IQR	Average Rank	Me	IQR	Z	*p*	r
Abacus (nr of beads per 30 s)	33.78	29.00	20.50	26.09	23.00	9.50	−1.72	0.085	0.22
Wall (nr of points)	32.98	3.00	1.00	26.91	2.00	2.00	−1.46	0.145	0.19
Vertical fishing-task completion (%)	32.43	91.00	21.25	27.48	78.00	32.50	−1.13	0.257	0.15
Vertical fishing-time (s)	23.27	45.00	27.50	36.97	66.00	60.50	−3.06	0.002	0.40
Vertical fishing-hand movement path coefficient	29.72	1.98	0.53	30.29	1.86	0.96	−0.13	0.897	0.02
Horizontal fishing-task completion (%)	31.28	46.50	30.75	28.67	45.00	29.50	−0.58	0.559	0.08
Horizontal fishing-time (s)	24.58	82.50	30.50	35.60	95.00	24.00	−2.46	0.014	0.32
Horizontal fishing-hand movement path coefficient	25.93	2.10	1.19	34.21	2.88	1.48	−1.85	0.064	0.24
Reaction time-task execution (%)	33.52	100.00	0.00	26.36	100.00	100.00	−2.57	0.010	0.33
Reaction time-time (s)	33.13	152.50	44.25	26.76	122.00	92.50	−1.43	0.154	0.19

Legend: me—median, IQR—quartile range, Z—standardized statistics of the Mann–Whitney U test, *p*—test probability, r—effect size.

**Table 4 sensors-22-04336-t004:** Comparison of the measurement before and after the stabilizing exercises in the study group.

	Study Group						
	Before		After				
	Me	IQR	Me	IQR	Z	*p*	r
Abacus (nr of beads per 30 s)	29.00	20.50	43.00	17.00	−4.44	<0.001	0.57
Wall (nr of points)	3.00	1.00	3.00	0.00	−3.22	0.001	0.42
Vertical fishing-task completion (%)	91.00	21.25	100.00	0.00	−3.41	0.001	0.44
Vertical fishing-time (s)	45.00	27.50	29.50	18.50	−3.65	<0.001	0.47
Vertical fishing-hand movement path coefficient	1.98	0.53	1.61	0.75	−3.29	0.001	0.42
Horizontal fishing-task completion (%)	46.50	30.75	66.00	56.00	−3.03	0.002	0.39
Horizontal fishing-time (s)	82.50	30.50	82.50	36.75	−1.34	0.180	0.17
Horizontal fishing-hand movement path coefficient	2.10	1.19	2.07	1.70	−1.14	0.254	0.15
Reaction time-task execution (%)	100.00	0.00	100.00	0.00	−1.00	0.317	0.13
Reaction time-time (s)	152.50	44.25	121.00	29.00	−4.05	<0.001	0.52

Legend: me—median, IQR—quartile range, Z—standardized statistics of the Mann–Whitney U test, *p*—test probability, r—effect size.

**Table 5 sensors-22-04336-t005:** Comparison of the measurement before and after exercise in the control group.

	Control Group						
	Before		After				
	Me	IQR	Me	IQR	Z	*p*	r
Abacus (nr of beads per 30 s)	23.00	9.50	28.00	14.00	−4.24	<0.001	0.56
Wall (nr of points)	2.00	2.00	3.00	1.00	−3.42	0.001	0.45
Vertical fishing-task completion (%)	78.00	32.50	100.00	9.00	−3.48	<0.001	0.46
Vertical fishing-time (s)	66.00	60.50	57.00	49.50	−3.30	0.001	0.43
Vertical fishing-hand movement path coefficient	1.86	0.96	1.67	0.69	−2.48	0.013	0.33
Horizontal fishing-task completion (%)	45.00	29.50	75.00	45.50	−3.28	0.001	0.43
Horizontal fishing-time (s)	95.00	24.00	89.00	30.50	−2.97	0.003	0.39
Horizontal fishing-hand movement path coefficient	2.88	1.48	2.63	1.71	−2.02	0.044	0.26
Reaction time-task execution (%)	100.00	100.00	100.00	0.00	−1.41	0.157	0.19
Reaction time-time (s)	122.00	92.50	113.00	61.50	−4.37	<0.001	0.57

Legend: me—median, IQR—quartile range, Z—standardized statistics of the Mann–Whitney U test, *p*—test probability, r—effect size.

**Table 6 sensors-22-04336-t006:** Comparison of people from the study and control groups for the measurement after exercise.

After	Study (n = 30)			Control (n = 30)					
	Average Rank	Me	IQR	Average Rank	Me	IQR	Z	*p*	r
Abacus (nr of beads per 30 s)	39.07	43.00	17.00	20.62	28.00	14.00	−4.13	<0.001	0.54
Wall (nr of points)	35.20	3.00	0.00	24.62	3.00	1.00	−2.95	0.003	0.38
Vertical fishing-task completion (%)	33.68	100.00	0.00	26.19	100.00	9.00	−2.10	0.036	0.27
Vertical fishing-time (s)	21.10	29.50	18.50	39.21	57.00	49.50	−4.05	<0.001	0.53
Vertical fishing-hand movement path coefficient	27.18	1.61	0.75	32.91	1.67	0.69	−1.28	0.200	0.17
Horizontal fishing-task completion (%)	30.27	66.00	56.00	29.72	75.00	45.50	−0.12	0.903	0.02
Horizontal fishing-time (s)	27.38	82.50	36.75	32.71	89.00	30.50	−1.19	0.234	0.16
Horizontal fishing-hand movement path coefficient	26.88	2.07	1.70	33.22	2.63	1.71	−1.42	0.156	0.18
Reaction time-task execution (%)	33.00	100.00	0.00	26.90	100.00	0.00	−2.61	0.009	0.34
Reaction time-time (s)	33.72	121.00	29.00	26.16	113.00	61.50	−1.69	0.091	0.22

Legend: me—median, IQR—quartile range, Z—standardized statistics of the Mann–Whitney U test, *p*—test probability, r—effect size.

## Data Availability

The data are available on request from corresponding author.

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
