# Peer review of "The Use of Armeo®Spring Device to Assess the Effect of Trunk Stabilization Exercises on the Functional Capabilities of the Upper Limb—An Observational Study of Patients after Stroke"

_sensors, 2022, doi:10.3390/s22124336_

Round 1

Reviewer 1 Report

This paper presents that the trunk stabilization exercises according to NDT Bobath have positive impact for the functional improvement of the affected upper limb in stroke patients. Sixty stroke patients are randomly recruited and divided into two groups, which are carried out with different therapies. The coordination of the upper limb is assessed based on three evaluation programs: "Vertical Fishing", "Horizontal Fishing", "Reaction time" and two proprietary tests: "Wall" and "Abacus". I unfortunately have several concerns with this paper.

1) The quality of English writing is poor. The words are not consistent. For example, the "Vertical Fishing" and "Horizontal Fishing" are expressed with "perpendicular hunting" and "horizontal hunting" in the abstract. Sometimes 'study groups' and sometimes 'test groups' are used in the content. Besides, some words are misspelled. For example, 'angular movementa' (Page 4, line 178), 'physiotherapist cheched' (Page 7, line 244). A lot of occurrences of this will make the paper difficult to understand. 

2) The format of the table should be consistent, therefore, suggesting that Table 1 and 2 use Three-line Table instead. Figure 3 is not regular, for example,  the rectangle is not closed, and the arrow is messy.  There are often two Spaces between two words in the content. Page 6, line 205, the paragraph format of "Vertical Fishing" is inconsistent with the "Horizontal Fishing " and "Reaction time". There are some question marks between two words in Page 3. 

3) The authors can consider placing the content of Armeo Spring assessment parameters (Page 7) within the content of Evaluation games (Page 6).

4) The introduction of the paper can be improved to motivate the problem better. Why the sudden transition from stroke (first paragraph in the intro) to trunk (the second paragraph in the intro). Some details about Outcomes and estimation (3.2) can be included in the intro.

5)  The study group was based on the use of the NDT Bobath concept, and the control group was based on the use of classic exercises. The results show that the study group can do better than the control group in upper limb coordination. In addition to the stable exercises of the trunk, Bobath therapy also is different from classic therapy in other factors. Therefore, how to prove that the upper limb coordination is better because of the factor of trunk stability exercises.

6) Please explain the meaning of acronyms ('Me', etc) in the Table.

Author Response

Manuscript ID: sensors-1730080

Title: “The use of Armeo®Spring device to assess the effect of trunk stabilization exercises on the functional capabilities of the upper limb – an observational study of patients after stroke.”

Dear Reviewers,

     Thank you very much for the analysis of our manuscript. We really appreciate your comments and indication of fragments that should be corrected and explained. Considering your suggestions, all mistakes were corrected. In order to avoid misunderstandings, changes introduced in the text are marked in blue and additionally, the manuscript was sent in the change tracking mode.

Reviewer #1:

Thank you very much for the very quick and thorough analysis of our manuscript.

The following comments and answers:

Comments and Suggestions for Authors

1) The quality of English writing is poor. The words are not consistent. For example, the "Vertical Fishing" and "Horizontal Fishing" are expressed with "perpendicular hunting" and "horizontal hunting" in the abstract. Sometimes 'study groups' and sometimes 'test groups' are used in the content. Besides, some words are misspelled. For example, 'angular movementa' (Page 4, line 178), 'physiotherapist cheched' (Page 7, line 244). A lot of occurrences of this will make the paper difficult to understand.

I actually used different words in the abstract, instead of fishing, I wrote hunting. Of course, I corrected these words. I also checked the entire text for similar mistakes. I also unified the term study group instead of the test group throughout the work. I corrected the misspelled words on page 4 verse 178 and page 7 verse 244. The entire manuscript was checked for linguistic errors.

Thank you very much for a comprehensive analysis of our manuscript.

2) The format of the table should be consistent, therefore, suggesting that Table 1 and 2 use Three-line Table instead. Figure 3 is not regular, for example,  the rectangle is not closed, and the arrow is messy.  There are often two Spaces between two words in the content. Page 6, line 205, the paragraph format of "Vertical Fishing" is inconsistent with the "Horizontal Fishing " and "Reaction time". There are some question marks between two words in Page 3.

I am not sure if I understood correctly the reviewer's suggestion. I have corrected Tables 1 and 2. In the present form, these are three columns. Indeed, the three-column layout seems to be more legible.

I also corrected figure 3 and I hope it is well done.

Indeed, there are often two or more spaces in the text of the manuscript. Of course, I reviewed and corrected the manuscript text in this regard.

I made a correction on page 6. The paragraph format is now the same.

I don't know why question marks appeared between the words. Of course, I made the appropriate correction on page 3.

Thank you very much for your careful study of our manuscript.

Thank you for these comments.

3) The authors can consider placing the content of Armeo Spring assessment parameters (Page 7) within the content of Evaluation games (Page 6).

With all due respect to all knowledge, time, and comments from the reviewer, we would like to leave the content of the parameters where they were originally written. The authors created the subsection "2.3. Devices and tests used at work" in order to describe in detail, inter alia, the technical aspects of the devices used.

Thank you very much for your comment.

4) The introduction of the paper can be improved to motivate the problem better. Why the sudden transition from stroke (first paragraph in the intro) to trunk (the second paragraph in the intro). Some details about Outcomes and estimation (3.2) can be included in the intro.

Indeed, the beginning of the introduction may seem divergent. I tried to improve the information contained therein a bit, which I hope will make it easier to understand the relationship.

With all due respect to the reviewer's comments, subsection 3.2. We considered outcomes and estimation complete, where the analysis is performed in accordance with the research questions posed. We could move the research questions to the introduction, but we decided that this way of presentation is more legible.

Thank you very much for your careful study of our manuscript.

Thank you for this comment.

5)  The study group was based on the use of the NDT Bobath concept, and the control group was based on the use of classic exercises. The results show that the study group can do better than the control group in upper limb coordination. In addition to the stable exercises of the trunk, Bobath therapy also is different from classic therapy in other factors. Therefore, how to prove that the upper limb coordination is better because of the factor of trunk stability exercises.

     The concept of NDT Bobath puts emphasis on the stabilization of the torso, explaining how it is very important for the proper functioning of the human body. At the same time, working with the use of the NDT Bobath concept helps to increase stability or improve equivalent patterns, which translates into the functioning of patients. In classical therapy, the proposed exercises affect the body frequently, locally. Step by step, we work out the improvement of specific body structures and their functions. The NDT Bobath concept seems to work comprehensively, so you can get good results faster. In our work, we examined patients at the beginning and after 10 days of therapy, and we obtained better results in the group in which the exercises were based on the NDT Bobath concept.

Thank you very much for your comment.

6) Please explain the meaning of acronyms ('Me', etc) in the Table.

The symbol Me stands for the median. To avoid misunderstanding, under each table I added a legend explaining the symbols. Thank you for this comment.

If any ambiguities or errors are still included, please feel free to give some tips so that we can write as correctly as possible.

Thank you very much for your insightful analysis, contribution and time.

Yours faithfully.

Reviewer 2 Report

The authors have carried out multiple studies using a well-known commercially available rehabilitation upper limb device on a group of 60 stoke patients. The paper doesn’t have any novelty aspects regarding the aims and scope of the proposed journal, as it mostly just an observational study applied on multiple patients using a randomized flow of participants through each stage of the study.

The significance of the proposed paper is important as it is based on a well-known commercially available rehabilitation upper limb device (Armeo Spring) paired with the device predefined three evaluation programs. The results revealed that the post-treatment analysis showed significantly better results with a significantly higher percent of task completion and better reaction times.  

The authors finding highlight the advantages of post-treatment use of the Armeo Spring device to helps stroke patients to recover over the usual classic exercises. Unfortunately, these findings are based on a commercially available rehabilitation device paired with the three predefine evaluation programs.

The research paper has a high interest for readers and scientists that are interested in developing rehabilitation equipment. But the paper is not suited for the “Intelligent Sensors” section of Sensors section.

Author Response

Manuscript ID: sensors-1730080

Title: “The use of Armeo®Spring device to assess the effect of trunk stabilization exercises on the functional capabilities of the upper limb – an observational study of patients after stroke.”

Dear Reviewers,

     Thank you very much for the analysis of our manuscript. We really appreciate your comments and indication of fragments that should be corrected and explained. Considering your suggestions, all mistakes were corrected. In order to avoid misunderstandings, changes introduced in the text are marked in blue and additionally, the manuscript was sent in the change tracking mode.

Reviewer #2:

Thank you very much for the very quick and thorough analysis of our manuscript.

The following comments and answers:

Comments and Suggestions for Authors

The authors have carried out multiple studies using a well-known commercially available rehabilitation upper limb device on a group of 60 stoke patients. The paper doesn’t have any novelty aspects regarding the aims and scope of the proposed journal, as it mostly just an observational study applied on multiple patients using a randomized flow of participants through each stage of the study.

Thank you very much for your insightful analysis, contribution and time.

The significance of the proposed paper is important as it is based on a well-known commercially available rehabilitation upper limb device (Armeo Spring) paired with the device predefined three evaluation programs. The results revealed that the post-treatment analysis showed significantly better results with a significantly higher percent of task completion and better reaction times. 

Thank you very much for the good evaluation of our work.

The authors finding highlight the advantages of post-treatment use of the Armeo Spring device to helps stroke patients to recover over the usual classic exercises. Unfortunately, these findings are based on a commercially available rehabilitation device paired with the three predefine evaluation programs.

Perhaps at this point, we were not understood by the reviewer. The Armeo® Spring device was used in our work to evaluate the progress of therapy, not the therapy itself. To evaluate the patients, we used evaluation games such as: "Vertical Fishing", "Horizontal Fishing" and "Reaction time". On the other hand, the therapy in the groups of patients was based on classical physiotherapy in the control group, and the therapy was based on the NDT Bobath concept in the study group. We were interested in whether the applied exercises improving the stability of the trunk (NDT Bobath) are of significant importance for the improvement of the function of the affected upper limb. Our research has proven this, although we recognized the limitations of our work.

Thank you very much for your careful study of our manuscript.

The research paper has a high interest for readers and scientists that are interested in developing rehabilitation equipment. But the paper is not suited for the “Intelligent Sensors” section of Sensors section.

Yes, indeed the manuscript may not fit this section of the journal. We received a proposal to submit a manuscript to this section as a thank you for the manuscripts I reviewed. Unfortunately, I did not have any other more relevant material at the moment, which is why the analyzed manuscript describes such issues. However, I hope that it can be published in the journal Sensors, perhaps in a different section, which I leave to the decision of the journal's reviewers and editors.

Thank you very much for your time.

Yours faithfully